# RPA-CRISPR/Cas12a-Based Detection of *Haemophilus parasuis*

**DOI:** 10.3390/ani13213317

**Published:** 2023-10-25

**Authors:** Kunli Zhang, Zeyi Sun, Keda Shi, Dongxia Yang, Zhibiao Bian, Yan Li, Hongchao Gou, Zhiyong Jiang, Nanling Yang, Pinpin Chu, Shaolun Zhai, Zhanyong Wei, Chunling Li

**Affiliations:** 1Institute of Animal Health, Guangdong Academy of Agricultural Sciences, Key Laboratory of Livestock Disease Prevention of Guangdong Province, Scientific Observation and Experiment Station of Veterinary Drugs and Diagnostic Techniques of Guangdong Province, Ministry of Agriculture and Rural Affairs, Guangzhou 510640, China; zkl06001@163.com (K.Z.); szy10252021@126.com (Z.S.); kedashi1994@163.com (K.S.); ydx10001@163.com (D.Y.); 15013267255@wo.cn (Z.B.); cyhlly@126.com (Y.L.); gouhc@hotmail.com (H.G.); jiangzy01@sina.com (Z.J.); 18165627183@126.com (N.Y.); cpp1900@163.com (P.C.); zhaishaolun@163.com (S.Z.); 2College of Veterinary Medicine, Henan Agricultural University, Zhengzhou 450046, China

**Keywords:** *Haemophilus parasuis*, RPA, CRISPR/Cas12a, detection

## Abstract

**Simple Summary:**

*Haemophilus parasuis* (*H. parasuis*, HPS) is a prominent pathogenic bacterium in pig production. Its infection leads to widespread fibrinous inflammation in various pig tissues and organs, often in conjunction with various respiratory virus infections, and leads to substantial economic losses in the pig industry. In this study, we used recombinase polymerase amplification (RPA) and clustered regularly interspaced short palindromic repeats (CRISPR) technology to establish a convenient detection and analysis system for *H. parasuis*. The process from sample to results can be completed within 1 h with high sensitivity (0.163 pg/μL of DNA template), which is 104 folds higher than the common PCR method. The specificity test results show that the RPA-CRISPR/Cas12a analysis of *H. parasuis* did not react with other common pig pathogens. Moreover, the method allows results to be visualized with blue light. The accurate and portable detection method holds great potential for *H. parasuis* control in the pig industry, especially in areas where specialized equipment is not available.

**Abstract:**

*Haemophilus parasuis* (*H. parasuis,* HPS) is a prominent pathogenic bacterium in pig production. Its infection leads to widespread fibrinous inflammation in various pig tissues and organs, often in conjunction with various respiratory virus infections, and leads to substantial economic losses in the pig industry. Therefore, the rapid diagnosis of this pathogen is of utmost importance. In this study, we used recombinase polymerase amplification (RPA) and clustered regularly interspaced short palindromic repeats (CRISPR) technology to establish a convenient detection and analysis system for *H. parasuis* that is fast to detect, easy to implement, and accurate to analyze, known as RPA-CRISPR/Cas12a analysis. The process from sample to results can be completed within 1 h with high sensitivity (0.163 pg/μL of DNA template, *p* < 0.05), which is 10^4^ -fold higher than the common PCR method. The specificity test results show that the RPA-CRISPR/Cas12a analysis of *H. parasuis* did not react with other common pig pathogens, including *Streptococcus suis* type II and IX, *Actinobacillus pleuropneumoniae*, *Escherichia coli*, *Salmonella*, *Streptococcus suis*, and *Staphylococcus aureus* (*p* < 0.0001). The RPA-CRISPR/Cas12a assay was applied to 15 serotypes of *H. parasuis* clinical samples through crude extraction of nucleic acid by boiling method, and all of the samples were successfully identified. It greatly reduces the time and cost of nucleic acid extraction. Moreover, the method allows results to be visualized with blue light. The accurate and convenient detection method could be incorporated into a portable format as point-of-care (POC) diagnostics detection for *H. parasuis* at the field level.

## 1. Introduction

*Haemophilus parasuis* (*H. parasuis*, HPS) is a prevalent bacterial disease in modern livestock farming, affecting animals of all age groups. It is caused by *H. parasuis* and is characterized by clinical symptoms such as multiple pleuritis, arthritis, meningitis, respiratory distress, and elevated body temperature [1,2,3]. The disease can also be accompanied by other pathogens or secondary infections. *H. parasuis* is a Gram-negative bacterium that thrives in media containing serum and NAD, and it forms distinct satellite colonies when co-cultured with *Staphylococcus aureus* [4]. The bacterium is known to have 15 serotypes, with serotypes 4, 5, and 12 being the most prevalent, leading to significant economic losses in the farming industry [5]. Early detection of infectious pathogens is crucial for disease control. PCR and qPCR techniques have been widely used for the detection of bacterial pathogens. While these methods have certain requirements for detection conditions and equipment, rapid on-site pathogen detection is still challenging. Recombinase polymerase amplification (RPA) is an isothermal amplification technique that utilizes a recombinase, polymerase, single-stranded binding protein, and primers to amplify the target at a temperature range of 37–42 °C [6,7,8]. The recombinase protein is combined with RPA primers to form a recombinase-primer complex and scan the homologous sequence in the dsDNA template. Then the primers were inserted at the cognate site by the strand-displacement activity of the recombinase. The single-stranded binding protein is bound to the replaced template chain to prevent the inserted primer from dissociating from the template. Generally, amplification of the target DNA sequence by RPA can be obtained in less than 20 min.

Due to its mild reaction conditions, high amplification efficiency, and simplicity of equipment, RPA has been increasingly employed for the rapid detection of bacteria, fungi, viruses, and parasites. It is particularly suitable for field testing requirements [9,10,11].

Clustered regularly interspaced short palindromic repeats (CRISPR) and CRISPR-associated proteins (Cas) are found in the immune systems of archaea and bacteria. It functions by recording and degrading exogenous nucleic acids under the guidance of a sequence-specific RNA molecule [12,13,14]. Cas12a, Cas12b, Cas13a, Cas13b, and Cas14 have proven collateral cleavage activity after binding to their specific targets, which has been used to detect nucleic acids in the field of diagnosis. Among those Cas, Cas12a possesses trans-cleavage activity in addition to its CRISPR-derived RNA (crRNA)-directed recognition and cleavage of specific dsDNA targets [15]. The trans-cleavage activity is triggered by the hybridization of crRNA with complementary target sequences, leading to the cleavage of single-stranded DNA from non-target sequences [16,17]. Building upon this discovery, researchers have applied Cas12a to pathogen detection by designing reporter molecules consisting of single-stranded DNA combined with quenched and fluorescent moieties. In the presence of a target sequence in the test sample, crRNA binds to the target sequence, forming a complex that activates the nuclease activity of Cas12a to cleave the target sequence. The trans-cleavage activity then performs arbitrary cleavage of the reporter molecule, and the presence of a fluorescent signal or not determines the test result.

In recent years, there has been an increasing demand for rapid and convenient pathogen detection methods. As the concentration of the clinical sample is often not high enough, it is necessary to pre-amplify the sample before testing to improve the detection sensitivity. To address this need, the Doudna group has combined RPA with CRISPR assays for the detection of human papillomavirus and has established a DNA endonuclease-targeted CRISPR trans-reporter system [18,19,20]. Since then, CRISPR assays have been widely utilized for the detection of various pathogens, including SARS-CoV-2, *Staphylococcus aureus*, and parasites. In this study, we have integrated RPA and CRISPR/Cas12a detection systems to develop a novel and rapid molecular assay for the detection of *H. parasuis*. Firstly, we amplified the *H. parasuis* Omp P2 gene using RPA technology. Secondly, incubation of CRISPR RNA (crRNA) and Cas12 proteins to form a crRNA-Cas12 complex under 25 °C. And then, the crRNA was used to activate the CRISPR protein to detect the amplified products. Once crRNA recognized and bound the target sequences, Cas12a was active in cleaving the ssDNA. The cleaved ssDNA makes its reporter molecule produce fluorescence, which can be detected under blue light (Figure 1). In this study, the RPA/Cas12a detection assay showed the potential to detect *H. parasuis* with high sensitivity and specificity. It does not need expensive experimental equipment and can be deployed in the field for rapid *H. parasuis* point-of-care (POC) detection.

## 2. Materials and Methods

### 2.1. Bacterial Strain and Growth Conditions 

*H. parasuis* (serovars 1, 2, 3, 4, 5, 6, 7, 8, 9, 10, 11, 12, 13, 14, and 15; designated HPS1, HPS3, HPS4, HPS5, HPS6, HPS7, HPS8, HPS9, HPS11, HPS12, and HPS15), *Streptococcus suis* (serovars II IX, designated SS2, SS9), *Staphylococcus aureus*, *Actinobacillus pleuropneumoniae*, *Escherichia coli*, and *Salmonella* strains were provided by the Institute of Animal Health, Guangdong Academy of Agricultural Sciences. All of those bacteria were isolated from clinical samples in China. *H. parasuis* was cultured in TSB (Difco Laboratories, Detroit, MI, USA) supplemented with 1 mg/mL nicotinamide adenine dinucleotide (NAD) and 10% bovine serum. *Streptococcus suis* was cultured in TSB supplemented with 5% bovine serum. *Staphylococcus aureus*, *Actinobacillus pleuropneumoniae*, *Escherichia coli*, and *Salmonella* were cultured in TSB. Bacteria were cultured overnight at 37 °C with 200 rpm circular agitation. The harvest concentrate of bacteria is OD600, which reaches 0.7–0.9. The above bacteria were stored at −80 °C with 20% glycerol.

Cas12a protein and NEB buffer 2.1 were purchased from New England Biolabs (Ipswich, MA, USA); the Bacterial Genome Extraction Kit was purchased from Takara (Kyoto, Japan); and the gel imager (TANNON4100) was purchased from Shanghai Tannon Technology (Shanghai, China).

### 2.2. Bacterial Genome Extraction

Bacterial strains were frozen and cultured for 12 h as described in Section 2.1. The genomic DNA was extracted using the Bacterial Genome Extraction Kit (Takara, Japan) according to the manufacturer’s instructions. The fast DNA extraction method is as follows: (1) centrifuge 1 mL of bacterial culture at 12,000 rpm/min for 5 min to obtain bacterial clumps; (2) wash the bacterial clumps three times using PBS; (3) add 100 μL of deionized water to re-suspension precipitation; (4) boil the samples under 100 °C for 10 min and centrifuge for 5 min at 12,000 rpm/min for 1 min; the supernatant is crude genomic DNA. 

### 2.3. Primer Design and RPA Reactions 

The RPA primers targeting the ompP2 gene of *H. parasuis* were designed. The primer-designed principle followed the protocol of Twist-Dx (Maidenhead, UK). The primer pair sequence is as shown in Table 1. RPA reactions were performed according to the manufacturer’s protocol (Twist Amp^®^Basic, Maidenhead, England). In short, the reactions were performed in a total volume of 25 μL, comprising RPA enzymes. The lyophilized RPA enzymes were re-suspended in 10 μmol/L upstream and downstream primers, 29.5 μL Primer Free Rehydration Buffer, and ddH_2_O to a total volume of 45.5 μL. And then, we divided the total volume into two tubes. One reaction needs 1 μL of DNA template and 1.25 μL of Mgo Ac (280 mmol/L), respectively. All reactions were amplified at 39 °C for 20 min.

### 2.4. crRNA and ssDNA Reporter Design

The crRNA was designed to target the RPA-amplified product of the ompP2 gene sequence, which is adjacent to the TTTN protospacer adjacent motif (PAM) site. The crRNA with base-complementary pairing to the target sequence was designed according to the position of the PAM sequence. The ompP2 1# crRNA sequence was 5′-UAAUUUCUACUAAGUGUAGAUUACAACACCAUGACCACCAUCAA-3′ (the final selected crRNA), and the 2# crRNA sequence was 5′-crRNA UAAUUUCUACUAAGUGUAGAUAUGUUACUCCAAAAUCUGGCGTG-3′. Once crRNA recognizes the target sequences and activates the Cas12a protein, any single-stranded DNA can be cut to activate the ssDNA reporter and produce fluorescence. The ssDNA reporter sequence was 5′-FAM-TTATT-BHQ1-3′. ssDNA was synthesized and purified by Sangon Biotech (Shanghai, China), and crRNA was synthesized by Huzhou Hippo Biotech (Huzhou, China). 

### 2.5. Optimized Cas12a Detection Reactions

The concentrations of the ssDNA reporter molecule, Cas12a, and crRNA are key factors affecting the efficiency of the CRISPR-Cas12a reaction. The RPA product (3 μL) was added to 30 μL of the CRISPR-Cas12a reaction mixture. The optimal reaction conditions were selected by observing the fluorescence intensity. At the same time, the fluorescence intensity of the sample was collected at 30 min for the Cas12a reaction. The assay was performed using the Roche Light Cycle 480 instrument on the 465–510 wavelength channel. The concentrations of the ssDNA reporter molecule were optimized at 450 nmol/L, 400 nmol/L, 350 nmol/L, 300 nmol/L, 250 nmol/L, and 200 nmol/L in the CRISPR-Cas12a reaction mixture containing 1 μL Cas12a (1 μmol/L), 1 μL crRNA (1 μmol/L), and 3 μL 10 × NEB buffer 2.1. Then, the concentrations of Cas12a and crRNA were optimized with the optimal ssDNA concentration as the test concentration. The optimal shear efficiency concentration was determined by an orthogonal test with the different concentrations of Cas12a (100 nmol/L, 50 nmol/L, 30 nmol/L, and 15 nmol/L) and crRNA (100 nmol/L, 50 nmol/L, and 30 nmol/L), respectively. 

### 2.6. Specificity Test

The RPA-CRISPR/Cas12a HPS assay specificity was evaluated by testing DNA from *Haemophilus parasuis*, *Streptococcus suis* type II, IX, *Actinobacillus pleuropneumoniae*, *Escherichia coli*, *Salmonella*, and *Staphylococcus aureus*, while ddH_2_O served as the negative control. The DNA of bacteria was generated by the Bacterial Genome Extraction Kit (Takara, Japan). 

### 2.7. Sensitivity Test

The genomic DNA of HPS5 was used in the sensitivity analysis of the assay. Serial dilutions of genomic DNA were made from 163 ng/μL down to 16.3 fg/μL. The limit of detection was determined by detecting the fluorescent signal in the tube containing the least DNA. The diluted templates were also subjected to a conventional HPS PCR detection assay to compare the sensitivity of the two methods. The method of conventional HPS PCR detection assays was previously described [21]. The detailed reaction system is as follows: 2×Taq buffer 10 μL, Primer-F1 (TATCGGGAGATGAAAGAC)/F2(GTAATGTCTAAGGACTAG)/R (CCTCGCGGCTTCGTC) 1 μL, respectively, cDNA 1 μL, H_2_O 6 μL. Then, the assay was performed with the following program: pre-denaturation at 98 °C for 4 min, denaturation at 95 °C for 30 s, annealing at 55 °C for 40 s with 35 cycles, and extension at 72 °C for 1 min.

### 2.8. Applications of RPA-CRISPR/Cas12a System 

The DNA extraction from 15 HPS strains involved the following steps: (1) Centrifugation of 1 mL of bacterial solution at 12,000 rpm for 2 min at room temperature, followed by discarding the supernatant; (2) addition of 1 mL of PBS solution and resuspension, followed by centrifugation at 12,000 rpm for 1 min and discarding the supernatant; (3) addition of 100 μL of H_2_O and heating in a metal bath at 100 °C for 5 min; (4) removal of the supernatant and determination of concentration using an enzyme marker. The concentration was determined using an ELISA. DNA templates of the 15 strains were amplified using the optimized RPA-CRISPR/Cas12a system, and fluorescence was observed under blue light at the end of the reaction.

## 3. Results

### 3.1. Primer and crRNA Screening

As the sequences of oligonucleotides are crucial to a rapid and sensitive RPA, primers need to be screened to establish a preferred detection method. At the same time, crRNA cutting efficiency is one of the most important aspects affecting detection efficiency. Based on the detection of target nucleic acids by Cas12a, different crRNAs bind to different target sites to cut ssDNA reporter molecules. We tested 4 primer pairs and 2 crRNA combinations for the target gene (ompP2), using 163 pg/μL of DNA for each reaction, respectively. The primer pairs and crRNA were visualized in Figure 2A. To screen out the best combination of primer and crRNA set for the ompP2 gene, a crRNA (1# and 2#) was used against 5 primer pairs (F1R1, F2R1, F3R3, F3R4, and F4R4), respectively. The primer pair (F1R1) and crRNA (1#) with the best performance were identified and used in subsequent experiments (Figure 2A,B). 

### 3.2. Optimization Results of the Reaction System

The results obtained from different concentrations of ssDNA reporter molecules indicate a positive correlation between the fluorescence intensity of CRISPR/Cas12a and the concentration of ssDNA reporter molecules. Specifically, higher concentrations of the ssDNA reporter molecule correspond to higher fluorescence intensity. The strongest fluorescence intensity was observed at an ssDNA reporter molecule concentration of 450 nmol/L, and no faint fluorescence was observed in the negative control. Therefore, an ssDNA reporter molecule concentration of 450 nmol/L was selected as the reaction condition.

To optimize the concentration of Cas12a and crRNA, we measured the fluorescence values at wavelengths of 465–510 nm for 15 min at 37 °C. Based on the fluorescence values, the highest fluorescence values were obtained with the Cas12a protein concentration of 100 nmol/L and the crRNA concentration of 50 nmol/L (Figure 3). However, there was no significant difference compared to a Cas12a protein concentration of 50 nmol/L and a crRNA concentration of 50 nmol/L. Considering the cost of reagents, we selected the Cas12a protein concentration of 50 nmol/L and the crRNA concentration of 50 nmol/L as the test concentrations.

### 3.3. Results of the Specificity Test

The RPA-Cas12a assay was subsequently employed to amplify the DNA templates of HPS5, *Streptococcus suis* type II and IX, *Actinobacillus pleuropneumoniae*, *Escherichia coli*, *Salmonella*, and *Staphylococcus aureus*. The results in Figure 4 showed that only the HPS5 DNA template elicited a discernible fluorescence reaction, whereas no fluorescence reaction was discerned for the other bacteria by the naked eye. The fluorescence intensities of *Streptococcus suis* type II and IX, *Actinobacillus pleuropneumoniae*, *Escherichia coli*, *Salmonella*, and *Staphylococcus aureus* showed no significant differences from the negative control (*p* < 0.0001). This observation attests to the assay’s specificity.

### 3.4. Results of the Sensitivity Test

The HPS5 DNA template was diluted in a tenfold gradient at concentrations of 163 ng/μL, 16.3 ng/μL, 1.63 ng/μL, 163 pg/μL, 16.3 pg/μL, 1.63 pg/μL, 0.163 pg/μL, and 16.3 fg/μL in sequential succession. All of these experimental cohorts evinced a more pronounced fluorescence response utilizing the RPA-Cas12a methodology (Figure 5A,B). Although the fluorescence intensity of 0.163 pg/μL was feeble, fluorescence was still discernible in comparison to the negative control (Figure 5A). The statistical analysis result showed that a significant fluorescence signal was different from the negative control (*p* < 0.05) obtained when the HPS5 DNA template was ≥0.163 pg/μL (Figure 5B). Thereby establishing the method’s sensitivity at 0.163 pg/μL, while the sensitivity of the conventional PCR assay was a mere 1.63 ng/μL (Figure 5C). These findings manifested that the sensitivity of the RPA-Cas12a assay surpassed that of the conventional PCR assay by a factor of 10,000, thereby significantly enhancing the clinical detection accuracy of *H. parasuis*.

### 3.5. Results of the Application of the RPA-CRISPR/Cas12a Assay System

The extraction of bacterial nucleic acid constitutes the initial step in disease detection. In laboratory settings, bacterial nucleic acid kits may be employed and operated in accordance with their respective instructions. Although the quality of extracted nucleic acid with a commercial kit is relatively good, it still has several defects, such as the operation steps being cumbersome and time-consuming. Professional experimental equipment must be available and completed by professionals. Otherwise, it can easily lead to nucleic acid extraction failure, pollution, and other problems. To save detection costs and time, we tried to replace the kit with a simple nucleic acid extraction method. The extraction of the bacterial genome via the boiling method, as outlined in Section 2.8, entailed a total duration of 10 min. The outcomes of the RPA-CRISPR/Cas12a assay conducted on the extracted bacterial DNA from *H. parasuis* (serotypes 1/2/3/4/5/6/7/8/9/10/11/12/13/14/15) evinced the presence of a fluorescent signal under blue light at the culmination of the reaction for all crude DNA extracts, whereas no fluorescent signal was discerned for the negative controls (Figure 6).

## 4. Discussion

*H. parasuis* is a common pathogenic bacterium in the respiratory tract of pigs, which can cause widespread cellulose inflammation (also called Glässer’s disease) when the immunity of pigs is decreased. *H. parasuis* has fifteen serovars, and some clinical strains remain indistinguishable from serotypes. The vaccine’s cross-protection is poor among different serotypes. Hence, effective pathogenic detection is important for Glässer’s disease prevention and control. The methodologies employed for the detection of *H. parasuis* encompass bacterium isolation, PCR, qPCR, and loop-mediated isothermal amplification (LAMP) [22,23,24]. Bacterium isolation is the golden standard for *H. parasuis* diagnosis. The method needs professional laboratories, experienced inspectors, and at least several days. Presently, conventional PCR assays are the most frequently utilized laboratory technique to detect *H. parasuis*. However, they necessitate temperature-variable equipment and time-consuming agarose gel electrophoresis to validate the product outcomes, thereby engendering a laborious process. qPCR testing is more effective than conventional PCR, but it needs more expensive equipment and kits. Many kits require qualified laboratories and professional testing personnel. In recent years, LAMP assays have garnered extensive scrutiny due to their heightened sensitivity and specificity. Nonetheless, the reaction mandates the incorporation of numerous pairs of primers, and the addition of reagents is cumbersome and susceptible to inadvertent errors, thereby impinging upon the results [25]. Consequently, there exists a demand for the development of a rapid and uncomplicated test for the detection of *H. parasuis*. As a specific nucleic acid recognition tool, the CRISPR-Cas tool possesses high specificity and sensitivity. Detection methods based on the CRISPR-Cas system have been widely developed in recent years. The CRISPR-Cas system has been applied to the detection of SARS-CoV-2, trichomonas vaginalis, the swine fever virus, and even the plant pathogen Bursaphelenchus [26,27,28]. In this study, we established a convenient and highly sensitive RPA coupled with a CRISPR/Cas12a assay to rapidly detect *H. parasuis* under blue light for result visualization. The assay requires three steps: DNA extraction, RPA reaction, and CRISPR detection, which can be finished in 60 min (Figure 1). Most bacterial detection methods require a commercial kit to extract DNA, which takes about half an hour. In this study, crude DNA of *H. parasuis* extracted by the boiling method also fit the RPA-CRISPR/Cas12a detection assay. It greatly saves detection costs and time (Figure 6). The RPA is an isothermal nucleic acid amplification platform that requires no equipment and is less time-consuming than conventional PCR and qPCR [6]. In this study, an RPA reaction lasting 20 min can obtain a sufficient detection template. The major limiting factor in the CRISPR-Cas system is the design of effective crRNA, which facilitates target recognition, binding, and cleavage efficiency [29,30]. Hence, the combination of RPA amplification primer and crRNA is critical to the RPA-CRISPR/Cas12a reaction. In this study, we screen 5 pairs of primers and 2 crRNAs that target the ompP2 gene of *H. parasuis*. We found that the combination of F1R1 primer and 1#crRNA had the best detection effect on *H. parasuis* (Figure 2). The fluorophore-modified aptameric sensor is one of the key techniques in molecular diagnostics. Fluorescence in Black Hole Quencher 1 (FAM-BHQ1) was the most widely used. 5′-FAM-TTATT-BHQ1-3′ was the ssDNA reporter sequence used for Cas12a proteins. Once ssDNA was bound to the ribonucleoprotein complex formed by Cas12a-crRNA, cas12a enzyme activity was activated to proceed with collateral cleavage of ssDNA reporter, which in turn produced intense fluorescence [30]. In this study, we found that the strongest fluorescence intensity was observed at an ssDNA reporter molecule concentration of 450 nmol/L, and no faint fluorescence was observed in the negative control (Figure 3A). Moreover, the molar ratio of the Cas12a protein to crRNA is a crucial factor for efficient signal amplification [31]. With a comprehensive consideration of fluorescence intensity and the cost of reagents, we selected a Cas12a protein concentration of 50 nmol/L and a crRNA concentration of 50 nmol/L as the test concentrations (Figure 3B). Under these optimized conditions, the RPA-Cas12a detection accurately identified the principal clinically prevalent strains of *H. parasuis* (serotypes 5, HPS5), with a sensitivity surpassing that of conventional PCR by a factor of 10,000 (Figure 4).

The assay also exhibited marked specificity, as it failed to elicit fluorescence with *Streptococcus suis* type II and IX (SS-2, SS-9), *Actinobacillus pleuropneumoniae* (APP), *Escherichia coli* (*E. coli*), *Salmonella*, *Staphylococcus aureus* (*S. aureus*), and *Erysipelothrix rhusiopathiae* (*E. rhusiopathiae*) (Figure 4). In addition, our assay can still detect 15 serotypes of *H. parasuis,* including serotypes 1/2/3/4/5/6/7/8/9/10/11/12/13/14/15 (Figure 6). Bacterial gene sequences may be mutant due to drug and environmental reasons. The fluorescence intensity of RPA/Cas12a detection may be different between clinical isolate strains in application. However, this does not affect the determination of a positive result. From our results, the RPA-CRISPR/Cas12a-based assay for *H. parasuis* is simpler and more facile to execute than conventional PCR and can be implemented in clinical settings where temperature-variable instruments are unavailable, making it a prompt and convenient assay.

## 5. Conclusions

In this study, the RPA-CRISPR/Cas12a-based detection of *H. parasuis* exhibits specificity, high sensitivity, and a short detection time and can visually determine the results under blue light. The limit of detection of the RPA-Cas12a assay is 0.163 pg/μL, which is 10^4^-fold higher than the common PCR method. The detected assay will significantly enhance the clinical detection accuracy of *H. parasuis*. The convenience and speed of RPA-CRISPR/Cas12a in this study make it possible to be incorporated into a portable format as point-of-care (POC) diagnostics detection for *H. parasuis* at the field level.

## Figures and Tables

**Figure 1 animals-13-03317-f001:**
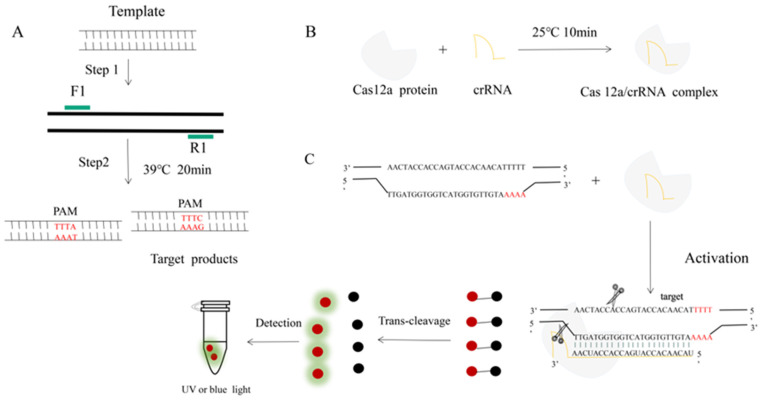
An RPA-CRISPR-Cas12a platform for the detection of *H. parasuis*. (**A**) Schematic illustration of the RPA. (**B**) Incubation of the amplified products (**A**) and Cas12 proteins to form a DNA-Cas12 complex under 25 °C for 10 min. Protospacer adjacent motif (PAM) sites guide the CRISPR/Cas12a-crRNA complex to recognize target sites to activate Cas12a effectors. (**C**) The crRNA recognized and bound the target sequences to activate the Cas12a protein. The activated Cas12a protein nonspecifically cleaves single-stranded DNA reporter molecules by trans-cleavage. The positive sample produces fluorescence and is detected by blue light.

**Figure 2 animals-13-03317-f002:**
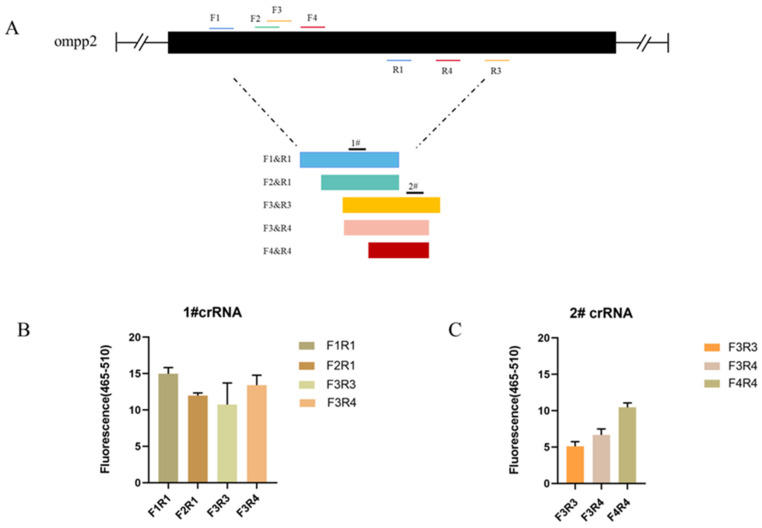
Optimization results of primer pairs and crRNA in the reaction system. (**A**) HPS ompP2 genome map showing primers and crRNAs. (**B**) Fluorescent results of 1# crRNA combined with 4 primer pairs (F1R1, F2R1, F3R3, and F3R4). (**C**) Fluorescent results of 2# crRNA combined with 3 primer pairs (F3R3, F3R4, and F4R4). Mean ± SD for 3 technical replicates for (**B**,**C**).

**Figure 3 animals-13-03317-f003:**
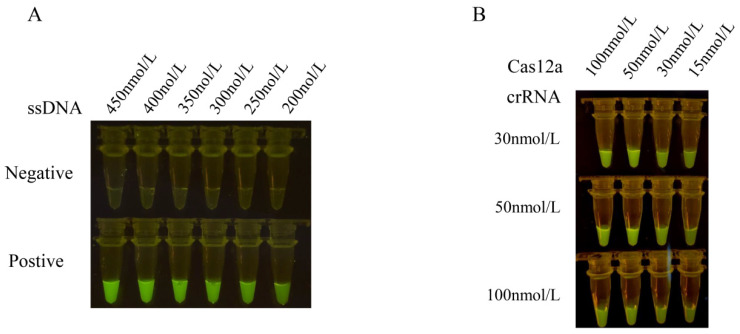
Optimization results of concentrations of ssDNA reporter, Cas12a and crRNA in the reaction system. (**A**) The concentrations of the ssDNA reporter molecule were optimized at 450 nmol/L, 400 nmol/L, 350 nmol/L, 300 nmol/L, 250 nmol/L, and 200 nmol/L in the CRISPR-Cas12a reaction mixture containing 1 μL Cas12a (1 μmol/L), 1 μL crRNA (1 μmol/L), and 3 μL 10 × NEB buffer 2.1. The results were observed under blue light. (**B**) The optimal shear efficiency concentration was determined by an orthogonal test with the different concentrations of Cas12a (100 nmol/L, 50 nmol/L, 30 nmol/L, and 15 nmol/L) and crRNA (100 nmol/L, 50 nmol/L, and 30 nmol/L), respectively. The concentration of Cas12a and crRNA was optimized with the optimal ssDNA concentration as the test concentration. The optimal reaction conditions were selected by observing the fluorescence intensity.

**Figure 4 animals-13-03317-f004:**
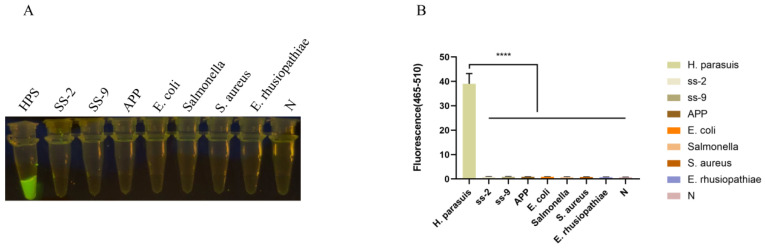
Specificity analysis. (**A**) Specificity assessment of the RPA-CRISPR/Cas12a-assay for the ompP2 gene of *H. parasuis*. Only the DNA from *H. parasuis* produced fluorescence signals, whereas DNAs from other pathogens and the negative control did not elicit discernible fluorescence signals. Seven swine bacterial pathogens, including *Streptococcus suis* type II and IX (SS-2, SS-9), *Actinobacillus pleuropneumoniae* (APP), *Escherichia coli* (*E. coli*), *Salmonella*, *Staphylococcus aureus* (*S. aureus*), and *Erysipelothrix rhusiopathiae* (*E. rhusiopathiae*), were evaluated. (**B**) Fluorescent results of different swine bacterial pathogens detected by the RPA-Cas12a assay were collected at 30 min for the Cas12a reaction. Three replicates were conducted for each test. Fluorescence intensity values are shown in the graph as mean ± SD. (N: negative control, ddH_2_O; **** *p* < 0.0001).

**Figure 5 animals-13-03317-f005:**
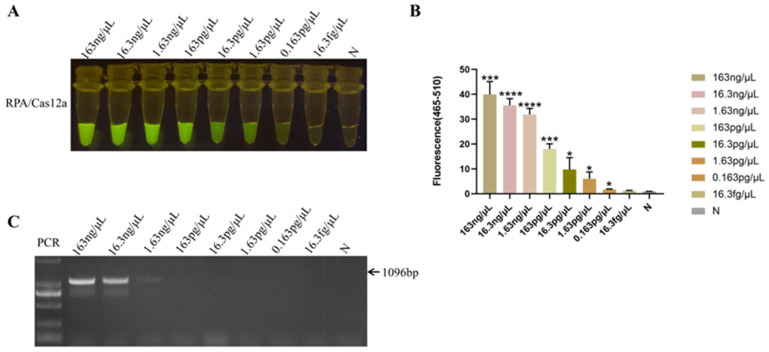
Sensitivity analysis. (**A**) Serial dilutions of HPS5 DNA for RPA-Cas12a limit of detection determination. The RPA product (3 μL) was added to 30 μL of the CRISPR-Cas12a reaction mixture for cleavage assays. The reaction results were observed in the fluorescence intensity under blue light. (**B**) The fluorescence intensity of each sample was collected at 30 min for the Cas12a reaction. Bar graphs represent fluorescent signals for the Cas12a reaction from A. Three replicates were conducted for each test. Fluorescence intensity values are shown in the graph as mean ± SD. (N: negative control, ddH_2_O; * *p* < 0.05, *** *p* < 0.001, **** *p* < 0.0001). (**C**) The serial dilutions of HPS5 DNA were amplified with specific primer sets in a classical PCR reaction. The amplification results were detected by agarose gel electrophoresis.

**Figure 6 animals-13-03317-f006:**
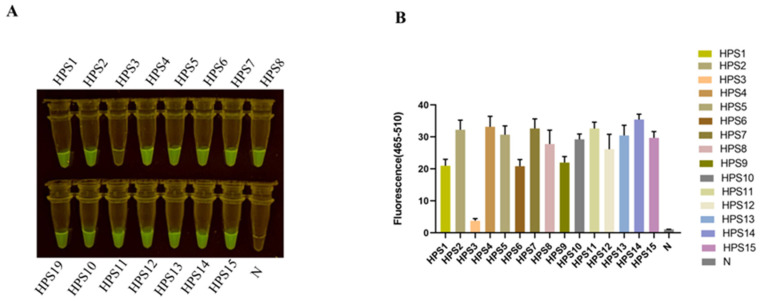
Detection of HPS clinical isolation strain. (**A**) The serotypes 1/2/3/4/5/6/7/8/9/10/11/12/13/14/15 of *H. parasuis* (HPS) were detected by the RPA-CRISPR/Cas12a assay. (**B**) Bar graphs representing fluorescent signals detected by the RPA-Cas12a assay were collected at 30 min for the Cas12a reaction. Fluorescence intensity values are shown in the graph as mean ± SD for 3 technical replicates.

**Table 1 animals-13-03317-t001:** The sequence of RPA primers.

Primer Name	Sequences 5′-3′	Note
F1	AATCTGTAACTGTGGCAGCAGGTTATACTCA	Selected primer pairs
R1	AGTACCATAAGAGTAGTTTCCATACACGCCA	
F2	GGTGTATACTTTGGTCTTAAATATGTCAACG	
F3	TAAGAAAGACAAAGATGGTGTATACTTTGGT	
R3	TGTCTTTGTTCTTGATTAAACGACCTTCAA	
F4	GTAGCTGTTGATGGTGGTCATGGTGTTGTAAA	
R4	CACCTAACATGAATTGATGAGCTGTTGCTT	

## Data Availability

The data are available upon request from the corresponding authors.

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
