# Peer review of "RPA-CRISPR/Cas12a-Based Detection of Haemophilus parasuis"

_animals, 2023, doi:10.3390/ani13213317_

Round 1

Reviewer 1 Report

The article brings novel procedure for the diagnosis of Haemophilus parasuis and might be a valuable tool to improve the control of Glasser disease, mostly because it is supposed to be a cheaper and quicker diagnostic method.

The language usage might be improved mostly on verbal tense and some awkward wording.

The author should address the following:

First, the pathogen is named Glasserella parasuis, not Haemophilus parasuis (it changed in 2020);

Second, the author needs to show that the method is suitable to all 15 sorovars (in the article the authors missed sorovars 2, 10, 13 and 14). Why these sorovars were not included in the study? Or, if they were tested, but not detected by the assay, it should be indicated by the authors. These sorovars are readily available in many labs around China and should be included in the study.

Third, what are the scale on figures? The authors need to explain the the scale of fluorescence on the grapha (Figures).

Fourth, discussion is poor. The authors should start discussion indicating their main finding and discuss the data afterwards considering the current diagnostic methos and the advantages or disadvantages of the novel method in comparison to existing methods.

Fifth, the conclusion sentence repeats the results. The author should make a real conclusion of the meaning of their work and the possible impact it has on the diagnosis of G. parasuis.

The article should be reviewed and improved prior to publication.

Can be improved. Please review tense usage and awkward wording.

Author Response

Thanks for your hard reviewing! We have answered all questions one by one and point by point. Please see the attachment.!

Reviewer 2 Report

Quick and simple diagnosis tools for pathogens such as Haemophilus parasuis are of high significance, for which the authors have developed a combination assay of an RPA and a CRISPR technology to facilitate rapid detection.

Overall the method developed is promising approach for saving cost, time and efforts; with possibility of using blue light for detection. The manuscript of scientific interest, however, it needs to revised for affirmative consideration.

Some comments:

1.     Authors should provide statistical information on SE/SD and p-values of the results in abstract and discuss it results section.

2.     Figure 1. Please show which one is template strand for crRNA binding.

3.     Authors could specify the sequences of crRNA, template DNA, probes, and activator or complentary DNA used for CRISPR assay design.

4.     Whether cas12a reaction performed at 25 0C or 37 0C (Fig 1A vs 1B)?

5.     What is the source of cas12a? which bacteria? LB or Fn?

6.     Please add a space between the unit and numbers, for example 1 nM (not 1nM). Follow this for all units. Revise Line 272…163ng/μL and elsewhere.

7.     Higher resolution figures are requested for all images and graphs.

8.     Line 334: Change QPCR to qPCR.

9.     In conclusions add statistical information, including LOD and provide implications of this assay for animal testing.

10.  I recommend authors to revise the relevant literature on CRISPR-Cas12a for diagnostics and discuss it in the manuscript. For example, see article: PMID: 36843874.

Minor editing is necessary.

Author Response

Thanks for your hard reviewing! We have answered all questions one by one and point by point. Please see the attachment!
